# Synthesis, Biological Activity, and Molecular Dynamics Study of Novel Series of a Trimethoprim Analogs as Multi-Targeted Compounds: Dihydrofolate Reductase (DHFR) Inhibitors and DNA-Binding Agents

**DOI:** 10.3390/ijms22073685

**Published:** 2021-04-01

**Authors:** Agnieszka Wróbel, Maciej Baradyn, Artur Ratkiewicz, Danuta Drozdowska

**Affiliations:** 1Department of Organic Chemistry, Medical University of Bialystok, 15-222 Bialystok, Poland; danuta.drozdowska@umb.edu.pl; 2Department of Physical Chemistry, Institute of Chemistry, University of Bialystok, 15-245 Bialystok, Poland; m.baradyn@uwb.edu.pl (M.B.); artrat@uwb.edu.pl (A.R.)

**Keywords:** trimethoprim, DHFR inhibitors, netropsin, MGBAs, drug design, molecular dynamics

## Abstract

Eighteen previously undescribed trimethoprim (TMP) analogs containing amide bonds (**1–18**) were synthesized and compared with TMP, methotrexate (MTX), and netropsin (NT). These compounds were designed as potential minor groove binding agents (MGBAs) and inhibitors of human dihydrofolate reductase (*h*DHFR). The all-new derivatives were obtained via solid phase synthesis using 4-nitrophenyl Wang resin. Data from the ethidium displacement test confirmed their DNA-binding capacity. Compounds **13–14** (49.89% and 43.85%) and **17–18** (41.68% and 42.99%) showed a higher binding affinity to pBR322 plasmid than NT. The possibility of binding in a minor groove as well as determination of association constants were performed using calf thymus DNA, T4 coliphage DNA, poly (dA-dT)^2^, and poly (dG-dC)^2^. With the exception of compounds **9** (IC50 = 56.05 µM) and **11** (IC50 = 55.32 µM), all of the compounds showed better inhibitory properties against *h*DHFR than standard, which confirms that the addition of the amide bond into the TMP structures increases affinity towards *h*DHFR. Derivatives **2**, **6**, **13**, **14**, and **16** were found to be the most potent *h*DHFR inhibitors. This molecular modelling study shows that they interact strongly with a catalytically important residue Glu-30.

## 1. Introduction

In recent years, folate metabolism has been recognized as an attractive and important target for the development of therapeutic agents in cancer therapy [1,2], as well as bacterial and parasitic infections [3]. Dihydrofolate reductase (DHFR) is a key enzyme that catalyses the NADPH-dependent reduction of 7,8-dihydrofolate (DHF) to 5,6,7,8-tetrahydrofolate (THF) in the reaction: DHF + NADPH + H+ → THF + NADP+. This compound is a precursor of the cofactors required for the biosynthesis of purine nucleotides, thymidine (precursor for DNA replication), and several amino acids such as glycine, methionine, serine, and *N*-formyl-methionyl tRNA [4,5,6,7,8]. DHFR inhibition causes partial depletion of intracellular reduced folates, which subsequently leads to limited cell growth [9]. Regarding the mechanism of action of antibacterial DHFR inhibitors, they block the synthesis of DNA, RNA, and proteins, causing cell growth arrest. As a result, DHFR becomes an important target for antimicrobial but also anticancer agents.

The most widely used human DHFR (*h*DHFR) inhibitor is the anti-cancer agent methotrexate (MTX) (Figure 1). However, it has been observed that MTX has various drawbacks, such as the development of resistance leading to loss of the active transport system through which MTX enters the cells [10,11].

Numerous reports show a variety of obtained structures in modifications of known DHFR inhibitors as potential anticancer agents. These compounds contain 1,3,5-triazine [12,13,14,15], 1,3-thiazole [16], 1,3,4-thiadiazole, and 1,2,4-triazole moieties in various fused heterocyclic systems [17,18,19]. Dihydrotriazine derivatives are also available in the literature [20]. Moreover, our last review presented the current state of knowledge on the modifications and structure–activity relationship of DHFR inhibitors as antitumor agents and showed that multitarget compounds are a promising approach for discovering new structures for anticancer therapy [21].

In turn, the most successful inhibitor of bacterial DHFR is trimethoprim (TMP) [2,4-diamino-5-(3,4,5-trimethoxybenzyl) pyrimidine] (Figure 1), which is a synthetic, broad-spectrum antimicrobial agent [22]. It is mainly used in the treatment of initial episodes of uncomplicated symptomatic urinary tract infections, both alone and in combination with a sulfonamide (e.g., sulfamethoxazole, sulfadiazine, sulfamoxole) [23]. TMP is a pyrimidine antifolate drug, which exerts antimicrobial activity by blocking the reduction of dihydrofolate to tetrahydrofolate, the active form of folic acid, by susceptible organisms [2,24,25]. One of our reviews presents an extensive range of research literature on the first and most recent achievements in TMP analogs as DHFR inhibitors and underlines new directions in developing and modeling DHFR inhibitors [26]. This literature analysis confirms that there are only a few reports showing the anticancer activity of TMP analogs. Singh et al. [27] modified the antibacterial agent TMP to compounds a and b (Figure 2), with promising anticancer applications. These two compounds had significant tumor growth inhibitory activities over 60 human tumor cell lines and exhibited appreciable interactions with DHFR [27]. Algul et al. [28] developed a new nonclassical series of propargyl-linked DHFR inhibitors. It was observed that interactions of propargyl-linked inhibitor (compound c) (Figure 2) with Leu22, Thr56, Ser59, Ile60 could potently inhibit human DHFR (*h*DHFR) in contrast to the weak inhibition of *h*DHFR by TMP.

In the field of cancer chemotherapy, designed multiple ligands (DMLs) offer several potential advantages over single target compounds, such as an increase of therapeutic efficacy or decrease of cancer drug resistance [28,29,30]. Our laboratory has an interest in the design and development of a novel class of TMP analogs both as DHFR inhibitors and DNA-binding agents, which are structurally related to netropsin (NT) and TMP (Figure 1). NT is a natural antibiotic, isolated for the first time by Finlay et al. from the *Streptomyces netropsis* strain [31]. This antibiotic is a classic representative of minor groove binding agents (MGBA). NT has been classified as an anticancer compound, forming non-intercalating bonds with DNA, but it is not used in medicine because of its high cytotoxicity [32].

The active site of all DHFRs in nature contains highly conserved glutamic acid residue (aspartic acid in bacteria) [33]. Although bacterial reductase has different amino acid sequences, mammalian DHFR shows remarkably similar inhibition profiles [34]. The catalytic activity of this residue is that it mediates the hydride transfer and protonation of dihydrofolate to tetrahydrofolate [35,36]. In the work of Wan et al. [37] the authors used neutron and ultrahigh-resolution X-ray crystallography to establish the catalytic activity of the *Escherichia coli DHFR* enzyme (*ec*DHFR). They deduced that Asp-27 (Glu-30 in mammalian DHFR) is responsible for binding the folate substrate in a favorable position for catalysis and that it is not protonated during this process, maintaining a negative electrostatic field in the active site. The importance of this residue was also extensively confirmed in a number of experimental and theoretical works [33,35,36,38,39,40]. Our derivatives aim to block this residue by forming a strong interaction and inhibiting DHFR enzymatic activity.

There is a significant number of molecular modelling studies concerning *h*DHFR inhibition as well as plenty of high resolution crystal structures of enzyme-inhibitor complexes [38,39,41,42]. This provides a solid foundation for structure-based design studies of potent *h*DHFR inhibitors. It is believed that the most relevant residues in the ligand binding to the *h*DHFR are Ile-7, Glu-30, Phe-31, Phe-34, Leu-67, Arg-70, and Val-115 [38,41,42]. These residues were also found to be important in our previous study [43]. Kerrigan et al. [44] have published a review on advances and applications of molecular dynamics simulations of dihydrofolate reductase. The authors concluded that “molecular mechanics calculations can work well to model the initial binding step of an inhibitor or substrate with DHFR”. However, to properly model the hydride transfer, one must perform costly QM calculations, as it involves the electron transfer process. An extensive theoretical study of structure-activity relationship of *h*DHFR inhibitors was reported by Tosso et al. [38]. In their work, the authors combined MD simulations with semiempirical, ab initio, and DFT calculations, providing an insight into the binding interactions of inhibitors from both structural and energetic points of view. Moreover, their selection of residues for reduced model is in great accordance with residues that were deemed important in our study.

In our previous study, we reported the design and synthesis of a novel series of TMP analogs (**A**–**F**) (Figure 3) containing amide bonds as novel DHFR inhibitors and candidates for antitumor drugs [43]. Compounds **B**–**C** and **E** were selected as the most active members of this study because they exhibited higher activity against the DHFR enzyme and higher binding affinity than standard TMP. Moreover, analogs **B,C**, and **E** were characterized by a higher binding strength to *p*BR322 plasmid. 

The determination of association constants values of drug–DNA complexes revealed that compounds **C** and **E** have high-value binding constants for T4 coliphage DNA, which confirms their minor-groove selectivity. The results obtained from the molecular modelling experiment showed that the introduction of an amide bond into the TMP analogs increases their affinity to human DHFR compared to unmodified TMP. In addition, in the first approach [43], it was hypothesized that compounds containing an amide bond and methylene bridge connecting two aromatic rings presented the highest decrease in fluorescence and showed the highest affinity to human DHFR. Additionally, it was found that even though our molecular docking studies showed a lower affinity for the **B**–**E** analogs, they were able to interact with the crucial residues Glu-30 and Phe-34. The effect of increasing the size of the aliphatic chain within the TMP analogs is not straightforward and requires further investigation. The binding energies of all analogs were significant and only about 1.2 kcal/mol lesser than the known DHFR inhibitor MTX (−9.5, kcal/mol) making these derivatives promising candidates for antimicrobial agents [43].

In view of these facts and in continuation of our previous efforts, this study reports solid phase synthesis of a new series of TMP analogs (**1–18**), accommodating amide bonds in the methylene bridge place and containing carbon–carbon double bonds and single bonds to explore the effect of conformational flexibility on activity against *h*DHFR. In addition, the pyrimidine ring was replaced by pyridine and benzene rings. Different groups, including electron-withdrawing (halogen) atoms, were substituted into the benzene ring to examine both their electronic and spatial effects on DHFR inhibitory activity and affinity to the active site of the enzyme. 

## 2. Results and Discussion

### 2.1. Compound Design and Synthesis

In our previous study, TMP analogs **B**–**D** displayed potent DHFR inhibitory activity with an IC_50_ value range of 0.72–0.99 µM [43]. Moreover, these compounds presented excellent DNA-binding activity expressed as % decrease of fluorescence, with values of 45.18% for compound **B** and 69.92% for compound **C**. In order to investigate structure–activity relationships (SAR) and to obtain more active DHFR inhibitors, **1**–**18** novel TMP derivatives were designed (Figure 4). The synthesis route of the target compounds is shown in Scheme 1, which was carried out according to the protocol presented earlier for netropsin analogs [45]. 

For the TMP derivatives preparation procedure, aromatic amino-nitro compounds **A_1–9_** and selected acid chlorides **E_1–5_** were used as substrates to obtain 18 novel TMP analogs containing an amide bond, as shown in Figure 4. Compounds with the structure **II** were obtained according to the reported procedure [46] from p-nitrophenyl carbonate Wang resin **I**, as shown in Scheme 1. After grafting the nitroamines to the resin, reduction of the nitro group of structure **II** was carried out using tin (II) chloride dihydrate in DMF. Acylation of **1**–**9** resin-bound amines **III**, using **E_1–5_** chlorides in the presence of DMAP in methylene chloride at room temperature produced the resin-bound nitro compounds with structure **IV**. Cleavage by 95% trifluoroacetic acid in dichloromethane gave a satisfactory yeld of the desired compounds with structure **V.** The structures of compounds A_1–9_, acid chlorides E_1–5_, the analytical and spectrometric data are presented in Appendix A.

### 2.2. Biological Activity: DNA-Binding Effects and Dihydrofolate Reductase (DHFR) Inhibition

The ethidium bromide assay showed that the investigated **TMP** analogs (**1–18**) can bind to plasmid DNA (Table 1). The results of this assay are presented as a percentage of the decrease in fluorescence of each substance in relation to the control, i.e., netropsin. The DNA-binding effect of NT in the same conditions was 74% [47,48]. The results revealed that all of the newly obtained compounds presented a DNA-binding effect (<100%). Significantly, all of the synthesized derivatives presented a higher degree of decrease of fluorescence than standard NT, with the exception of compounds **1**, **4**, **5**, and **10**, where the percentage decrease of fluorescence was 74.54%, 87.99%, 72.56%, and 70.11%, respectively. In addition, compounds **1**, **5**, and **10** showed a similar degree of DNA binding to NT. Table 1 presents that compounds **13–14** (49.9% and 43.85%) and **17–18** (41.68% and 42.99%) had a higher binding affinity to pBR322 plasmid compared to the compounds **A**–**F** synthesized earlier. The values of the association constants demonstrated that all the compounds can bind to the studied DNAs. The affinity constants of compounds **1–18** in the range of 1.2–5.8 × 10^5^ M^−1^ indicated moderate interactions with calf thymus.

The values of binding constants in the range of 0.7–6.4 × 10^5^ M^−1^ for T4 coliphage DNA for derivatives **1–18** was evidence of their minor-groove selectivity, because the major groove of T4 coliphage DNA is blocked by a-glycosylation of the 5-(hydroxymethyl)cytidine residues [49]. These data indicated that compounds **1–18** had interacted with AT as well as GC-base pairs and we can observe the greatest preference for AT-base pairs of compound **14** and for GC-pairs of **3**. All of the compounds bound to AT-rich sequences weaker than netropsin but some of them bound stronger to GC-rich sequences, e.g., compounds **3**, **6**, and **10**. Compound **18** indicated the highest value of affinity constants for T4 DNA (6.4 × 10^5^ M^−1^) as well as for calf thymus DNA (5.8 × 10^5^ M^−1^). This suggests that this compound is the best minor groove binding agent, although without significant selectivity (AT−4.5 × 10^5^ M^−1^, GC−4.6 × 10^5^ M^−1^). As can be seen from Table 1, all of the TMP (**1–18**) analogs, except **9** (IC_50_ = 56.05 µM) and **11** (IC_50_ = 55.32 µM), showed better inhibitory properties against *h*DHFR than standard TMP, with IC_50_ values from 0.89 to 30.02 µM, but neither of the presented derivatives was more active than the MTX.

Seven derivatives (**2**, **6**, **13–14**, and **16–18**) showed the best inhibitory properties against *h*DHFR with IC_50_ values ranging from 0.88 to 2.09 µM. These biological results could be analyzed on the basis of the type of ring (benzene, pyrimidine, pyridine), position on the benzene ring, the nature of the substituents: -I, -Cl, -F, -NH_2_, -OCH_3_, and the influence of the length of the carbon chain between the two aromatic rings. All of the synthesized TMP analogs have an amide bond incorporated between two aromatic rings. Compound **16** bearing an amide and carbon–carbon double bond connecting the benzene ring (4′-Cl, 3′-NH2) and 3′,4′,5′-trimethoxybenzene ring exhibited the best inhibitory activity with IC_50_ = 0.88 µM. Compounds **13** (IC50 = 0.89 µM), **14** (IC50 = 0.97 µM), **17** (IC50 = 1.22 µM), and **18** (IC50 = 2.09 µM), also containing carbon–carbon double bonds connecting two aromatic rings, presented similar behavior in biological activity. In addition, similar biological results were obtained for analog **2**, bearing a benzene ring (4′-F, 3′-NH2) and **6** with a pyridine ring connected by an amide bond in place of a methylene bridge to the 3′,4′,5′-trimethoxybenzene ring, with IC_50_ = 1.11 µM and 0.99 µM, respectively.

The introduction of a chlorine substituent in position 3′ of the benzene ring (compound **4**; IC_50_ = 5.02 µM) did not increase the *h*DHFR inhibiting activity but seemed to decrease it, when it was compared to compound **16** (4′-Cl-substituted benzene ring). TMP derivatives with 3′,5′-dimethoxy-; 3′4′-dimethoxy-, or 3′-methoxybenzene in the structure (compounds **8**, **9**, **10**, and **11**) were significantly less active (IC_50_ = 10.59, 56.05, 19.20, and 55.32 µM, respectively) than those with built-in 3′,4′,5′-trimethoxybenzene, with the exception of compound **7**, which showed better activity (IC_50_ = 3.02 µM) than these compounds.

### 2.3. Molecular Docking

The values of binding energy for ligands **1–18** and DHFR from our molecular docking studies are presented in Table 1. We tested these molecules against typical DHFR inhibitors: MTX and NT, as well as the TMP and its analogs **A**–**F** from our previous work [43]. The results suggest that our new set of compounds show significant affinity towards *h*DHFR.

Derivatives **2**, **6**, **13**, **14**, and **16** were found to be the most potent inhibitors, both in the experiment and the molecular docking study (Table 1). Therefore, further analysis will concern only these molecules. It is worth mentioning that binding modes, concerning molecules with the same molecular scaffold (**2**, **6** and **13**, **14**, **16**), are nearly identical and their interactions with residues in the protein’s active site are also very similar. These modes differ from one another by substituents and overlap with each other in our molecular docking study (Figure 5).

Tosso et al. [38] reported a series of *h*DHFR inhibitors with aromatic ring moieties substituted with the amine group. Their best binding modes show a substantial interaction with negatively charged glutamic acid 30 and amino group, further stabilized by π–π stacking with Phe-34. This is further validated by comparing the crystal structures of *h*DHFR with different inhibitors from X-ray diffraction experiments (Figure 6), where a common feature in binding poses can be noticed. The carboxylate group of Glu-30 acts as an anchoring point for the 2-amino group in inhibitors involving this kind of moiety. Results from our molecular docking study show that this is indeed the most favorable position for molecules **2** and **6**. For derivatives **13**, **14**, and **16**, however, the affinities associated with this kind of binding were ~0.5 kcal/mol higher than the best modes predicted by AutoDock Vina [50]. Because of the imperfections of molecular docking algorithms [50,51] and the previously discussed importance of interaction with Glu-30, we decided to use structures of **13**, **14**, and **16** where the amino group and aromatic ring are arranged in a similar way to the other inhibitors. The BIOVIA Discovery Studio software [52] was used to search for residues involved in binding the studied ligands to the receptor (Figure 7). A wide range of different types of interactions was observed, namely hydrogen bonds, interactions involving π orbitals, alkyl hydrophobic interactions, as well as a halogen bond in the case of molecule **2**.

Compounds **13**, **14**, and **16**, which are analogs with an additional double bond in the chain linking two aromatic rings, were found to have the highest affinity towards *h*DHFR among the tested molecules. They form conventional hydrogen bonds with Glu-30, Lys-55, and Thr-146. An interaction via t-shaped π stacking with Phe-34 is also observed, as well as a number of different hydrophobic interactions with residues Ile-7, Ala-9, Ile-16, Lys-55, and Val-115 (Figure 7a). These analogs were also found to form non-conventional hydrogen bonds with Thr-56, Gly-117, Thr-146. For molecule **14**, an additional interaction with Ile-7 involving a 3′-methoxy group in a 4-aminophenyl ring was detected. This molecule, along with **16**, was found to be the most potent inhibitor in this study, with a score of −8.2 kcal/mol.

Molecules **2** and **6** also had high binding affinities (−8.0 kcal/mol). These compounds form hydrogen bonds with Ser-59, Tyr-121, Thr-146, and either Glu-30 (**2**) or Ala-9 (**6**). There is also a π–π interaction between this group of ligands and residue Phe-34 (Figure 7a,b). Furthermore, an amide-π type of contact was observed between a ligand’s aromatic ring and the main chain of residue Val-8. For derivative **2** there is also a halogen bond between fluorine and Ile-7. Additionally, several different hydrophobic interactions were found with Ala-9, Ile-16, and Leu-22. The binding mode of molecule **6** does not seem to directly involve interaction with Glu-30, but the binding takes place in its very close vicinity. Our molecular dynamics calculations proved that strong contact is eventually formed.

The details of the docking studies for **A**–**F**, MTX, NT, and TMP can be found in our previous work [43], but for the sake of comparison, we have presented the necessary values in Table 1. Our newly synthesized TMP analogs exhibit a higher binding affinity than unmodified TMP (−7.5 kcal/mol for TMP compared to −8.2 kcal/mol for derivatives **14** and **16**). The modification that appears to affect binding energy the most is the addition of a double bond next to the amide group (**12–18**). All of the derivatives with double bonds, except for **12** and **15**, have a binding energy of −8.0 kcal/mol or better.

Another change introduced into our TMP analogs was the use of either a benzene, pyridine, or pyrimidine ring (Scheme 1). The best results were achieved for molecules with a benzene ring. This is reflected in the binding energies (Table 1), where eight out of the ten best binding modes consist of compounds with a benzene ring. On the other hand, molecules **10** and **12** with a pyrimidine ring are at the bottom of the energy ranking. Their performance in our inhibition experiment was also worse compared to the other derivatives. This suggests that the addition of nitrogen heteroatoms decreases binding affinity towards DHFR. This is in agreement with data from our work dealing with TMP analogs **A**–**F** [43], where we found that benzene containing inhibitors are also more potent.

The effect of fewer methoxy substituents in the trimethoxybenzene ring (molecules **6**–**11**) was tested. We found that the more methoxy groups connected to the ring the better the results. Methoxy groups are involved in many interactions within the active site pocket of the studied protein as a hydrogen bond acceptor. Moreover, many TMP derivatives showing considerable antimicrobial activity contain the trimethoxybenzene group [28,43,56]. The addition of halogen atoms as substituents (-F,-Cl,-I) in the benzene ring was studied as well, for molecules **1**–**5** and **16**–**18**. Since there was only one halogen bonding interaction observed (Figure 7a), their effect on the binding energy is considered negligible. The amide group plays an important role in binding of the ligand, but also elongates the chain between two aromatic rings, which makes interaction with positively charged Lys-55 possible. It is involved in numerous interactions with trimethoxybenzene rings (Figure 7c–e). Generally, it can be said that the results from molecular docking are in accordance with the experimental data. Two groups of derivatives had especially high inhibition activity: **13**, **14**, **16**, **17**, **18**, and **2**, **6**, which qualifies them as potential anticancer agents.

### 2.4. Molecular Dynamics

Derivatives **2**, **6**, **13**, **14**, and **16** proved to be the most potent inhibitors, both in the DHFR inhibition experiment and the molecular docking study. Hence, they were selected for further investigation. The results from Root Mean Square Deviation (RMSD) analysis showed that all the tested inhibitors have a tendency to lower this value compared to DHFR during the entire simulation, thus indicating that all of them have a stabilizing effect on *h*DHFR. Derivative **13** was found to have the greatest impact on protein stabilization, with RMSD oscillating around 1.1 Å for the whole 20 ns run. The only slight fluctuation was from 15–17 ns when it increased to 1.5 Å (Figure 8a). Compounds **2** and **6** exhibited the highest RMSD values during the first 10 ns, after that they stabilized at a level similar to the rest of the tested molecules. Derivative **14** showed growth in the RMSD value during the last 10 ns, which was also the case for **16**, but on a smaller scale. The RMSD value averaged over 20 ns was 1.994, 1.505, 1.395, 1.161, 1.483, and 1.394 for DHFR, **2**, **6**, **13**, **14**, and **16**, respectively.

A comparison of the Solvent Accessible Surface Area (SASA) of the investigated ligands suggests that all of them cause slight protein expansion (Figure 8b). Analogs **2** and **14** had the slightest impact on the protein surface area, and remaining analogs had a similar tendency as apo-protein for the first 12 ns. After this time SASA was higher and remained that way until the end of the simulation. Molecule **6** was determined to have the biggest effect on the enzyme, although the relative surface area expansion was only 2.6% when comparing the averaged values of **6** and DHFR. None of the inhibitors displayed protein compacting, except for very short intervals, when SASA for **2** and **14** was momentarily lower than that of DHFR. The averaged values of SASA in (Å^2^) for each system were 11,088 (DHFR), 11,202 (**2**), 11,377 (**6**), 11,322 (**13**), 11,285 (**14**), and 11,296 (**16**).

RMSF analysis helps to understand how the flexibility of particular regions of a macromolecule is affected by ligand binding. As expected, the lowest fluctuations were observed for residues creating *α* helices or *β*-sheets secondary structures. Derivative **2** was found to decrease fluctuations for almost all residues, except for 9–32 and 168–179, where the trend was the same as for DHFR. Another meaningful stabilization was noticed for molecule **16**, where RMSF decreased for regions 33–89 and 157–168, but slightly increased for 10–17 and 134–148. Compounds **6** and **14** caused higher fluctuations in regions 5–25, 114–121, 135–153, and 173–178, while at the same time decreased the RMSF value for regions 40–48, 100–110, and 159–168. However, molecule **6** was observed to have a greater effect on both lowering and elevating flexibility for most of these amino acids. Ligand **13** had similar fluctuations as DHFR for most of the protein sequence, with the small exception of 72–111 and 163–179, where RMSF was lower for the former and higher for the latter. In conclusion, we found residues Lys-18, Val-43, Lys-55, Lys-63, Lys-80, Arg-91, Lys-108, Lys-108, Lys-122, and Leu-153 to be flexible in all the researched cases. We noticed that these residues were mostly charged, located at the surface of the protein where they are exposed to the solvent and are not involved in forming the secondary structures (except for Lys-55 and Lys-122 which are part of α helices).

Radius of gyration (Rg) for DHFR and protein–ligand complexes showed small deviations (fraction of Å), indicating that the systems are tightly packed and do not undergo significant structural changes (Figure 8d). When compared to apo-protein, all analogs were observed to slightly increase Rg for the entire simulation, apart from **14**, which displayed smaller values during the first 9 ns. Compound **6** showed the highest Rg up until 10 ns, after which the Rg of derivative **2** started to increase considerably until the end of the simulation. Molecule **16** exhibited brief periods of decreased Rg around 13, 16, and 17–20 ns. The results averaged over the entire simulation are: 16.395 (DHFR), 16.594 (**2**), 16.590 (**6**), 16.531 (**13**), 16.446 (**14**), and 16.539 (**16**). Consequently, molecule **14** is expected to have slight advantage over the rest of the tested inhibitors, although all deviations from the Rg value of apo-enzyme are marginal.

The potency to form polar interactions for our derivatives was investigated by detecting the number of hydrogen bonds (H-bonds) between receptor and ligand during the simulation process. Because of the essential role of Glu-30 in the catalytic activity of *h*DHFR, we paid special attention to non covalent bonds involving this residue. Since glutamic acid is negatively charged, each of the oxygen atoms in the carboxyl group can serve as a potential H-bond acceptor. Moreover, we observed a strong interaction between derivative **2** and the main chain oxygen of Glu-30. The detailed analysis of H-bond occupancy (Table 2 and Figure 9) for each of our inhibitors, showed great affinity towards this residue for the majority of the MD simulation. In Figure 9, we can observe that, overall, compound **6** exhibits the highest number of H-bonds throughout the production run. Nonetheless, it is molecule **2** that formed form the most stable connection to the critical Glu-30 residue, as well as the widest range of different connections.

Derivative **16** interacted with the Glu-30 site roughly half of the simulation time, except for the interval between 2.5 ns and 12.5 ns, when it slightly moved away in favor of forming H-bonds with Ser-59 and Thr-56. In the case of molecule **14**, we can see that only during the last 3 ns interaction with Glu-30 is reduced. Other than that, our newly synthesized inhibitors showed significant affinity to the catalytically critical glutamic acid and were additionally stabilized by interactions with other residues.

Ala-9 acts as an H-bond donor, interacting via peptide N-H in the main chain with the oxygen atom in the amide group of our ligands. In the case of derivative **6**, Ala-9 was also briefly involved in interactions with nitrogen heteroatom of the pyridine ring through the main chain N-H group. Thr-56 was observed to form non-covalent interactions with threonine’s polar O-H as a donor and oxygen in the methoxy groups as an acceptor. A bond with residue Thr-136 was found only for molecule **2**, which was a consequence of fluorine interacting with the Thr-136 hydroxyl group. Amino acid Ser-118 participated in stabilizing the trimethoxybenzene moiety of the studied inhibitors mostly via main chain N-H, although the side chain O-H group was also involved but to a lesser extent. Visual analysis of MD trajectories showed that Phe-34 is positioned relative to the ligands in such a way that favors T-shaped π–π stacking. However, π–σ interaction between the phenylalanine and the inhibitor’s amino group was also present for derivative **6**. Since residue Val-115 had no donor or acceptor atoms in the side chain, an H-bond formed between the main chain carbonyl and ligand’s amide N-H. Short interactions with Tyr-121 were observed for **13** and **14**. Tyrosine’s O-H group acted as an H-bond acceptor, whereas amide N-H was a donor. Only molecule **16** bonded with Leu-27 during the simulation. This amino acid interacted with the inhibitor’s amino group and was both a donor and acceptor via its peptide N-H and C=O groups, respectively.

## 3. Materials and Methods

### 3.1. General Information

All reagents were purchased from Fluka (Sigma-Aldrich sp. z o.o., Poznań, Poland), Merck (Darmstadt, Germany), or Alfa Aesar (Karlsruhe, Germany), and used without further purification. Dichloromethane (DCM) and dimethylformamide (DMF) were stored in 4 Å molecular sieves. 1H-NMR and 13C-NMR spectra were recorded on a Bruker AC 400F spectrometer (Bruker corp., Fällanden, Switzerland) using TMS as the internal standard; chemical shifts were reported in ppm. Ethidium bromide was purchased from Carl Roth GmbH (Karlsruhe, Germany). Plasmid pBR322 was purchased from Fermentas Life Science (Vilnius, Lithuania). The Dihydrofolate Reductase Assay Kit was purchased from Sigma Aldrich (St Louis, Missouri, USA).

### 3.2. Synthesis

Solid-phase synthesis of the new compounds **1**–**18**, shown in Scheme 1, was carried out according to the protocol presented earlier for netropsin analogs [45].

#### Procedure for the Synthesis of Compounds **1**–**18**

Solid-phase synthesis of the new compounds **1**–**18**, shown in Scheme 1, was carried out according to the protocol presented earlier for NT analogs [46]. The preparation of TMP derivatives was performed with p-nitrophenyl Wang resin in the same way as shown for compound **I**. The resin (0.5 g; 0.41 mmol; 0.81 mmol/g) reacted with substrates **A_1_** (1.64 mmol), which were dissolved in DCM (10 mL) with the addition of pyridine (177.22 μL; 2.2 mmol), and then arranged in the parallel reaction vessels (**I**). Intermediates (**II**) were reduced by a solution of SnCl_2_ in DMF (1 M, 10 mL). The next stage of synthesis was acylation of the amine groups of (**III**) by using the substance **En_1_** (1.64 mmol). The acid chlorides **En_1_** were dissolved in DCM. The acid chlorides **En_1_** were dissolved in DCM. The coupling reactions were carried out overnight with the addition of a catalytic amount of DMAP (4-dimethylaminopyridine) at room temperature to produce the resin-bound compounds. Each resin-bound intermediate (**IV**) was washed before proceeding to the next stage. In the last stage of the process, the resins were dried and treated with TFA/DCM (50:50) [42]. After evaporation of the solvents, we yielded the products **1–18** as glaze solids. We tested the purity of the compounds by TLC (DCM:MOH:NH_3_ (7:2:1)). It was necessary to purify the obtained products, which was done by preparative chromatography using the same eluent. Their 1H and 13C NMR spectra in CD3OD were in agreement with the assigned structures and these data are provided. After evaporation of the solvents, we yielded the products **1**–**18** as glaze solids.

Their 1H and 13C NMR spectra were in agreement with the assigned structures and these data are provided (CD3OD). The structure, analytical, and spectrometric data are presented in Appendix A.

### 3.3. Ethidium Bromide Assay—DNA-Binding Effects

The effects of the investigated compounds **1**–**18** on plasmid *p*BR322 were determined in accordance with the procedure described previously [25,57]. Fluorescence readings are reported as % fluorescence relative to the controls.

### 3.4. Ethidium Displacement Bromide Assay—Determination of DNA-Binding Constants

The fluorescence of DNA solutions (calf thymus DNA, T4 coliphage DNA, poly(dA-dT)_2_ and poly(dG-dC)_2_ with the investigated compound (final concentrations 10, 50, 100 μM) was measured by fluorescence spectrophotometer Infinite M200 TECAN at room temperature according to the procedure described above. The determination of DNA-binding constants was described in previous articles. The apparent binding constant was calculated from KEtBr [EtBr] = Kapp [drug], where [drug] = the concentration of the tested compound at 50% reduction of fluorescence and KEtBr and [EtBr] are known [25,47]. The results are reported as a percentage of fluorescence decreases in Table 1. The compounds **1**–**18** and their DNA-bound complexes showed neither optical absorption nor fluorescence at 595 nm and did not interfere with the fluorescence of unbound ethidium bromide.

### 3.5. Dihydrofolate Reductase (DHFR) Inhibition Assay

The effects of novel TMP analogs (**1–18**) on the activity of recombinant human DHFR were determined by dihydrofolate reductase (DHFR) inhibition according to the reported methods and according to the instructions supplied with the set and recommended by the manufacturer [25,58]. The results are reported as IC_50_ (50% inhibition of enzymatic activity) in Table 1.

### 3.6. Molecular Docking

To examine the affinity of our series of novel inhibitors to *h*DHFR and to identify the most potent ones for further molecular dynamics simulation, we conducted a molecular docking study. The structure of the receptor protein was obtained from the Protein Data Bank (PDB: 1U72, resolution 2.0 Å) and prepared for calculations. The water molecules as well as the MTX ligand were removed and polar hydrogen atoms were added. The software used for the preparation of molecules and computation in this study was AutoDock Vina [39] (version 1.1.2, Scripps Research Institute, La Jolla, CA, USA). In our previous study [43], we tested a series of TMP **A**–**F** and performed a redocking study of MTX with the same protein structure used in this paper to validate our method. It proved to be appropriate for studying these derivatives. Hence, we used the same search space within the active site of *h*DHFR to find the best binding modes for molecules **1**–**18**.

### 3.7. Molecular Dynamics

To get better insight into how our inhibitors interact with *h*DHFR and how they affect its structure, we conducted a 20-ns molecular dynamics (MD) simulation using the NAMD software [59], with CHARMM36 force field for proteins [60] and the CGenFF Parameters [61] for small organic molecules. Initial parameters for all five ligands were assigned by the CGenFF program [62]. Partial atomic charges, as well as force field parameters for angles and dihedral angles, that were not assigned by analogy were all optimized using the Force Field Toolkit (ffTK) [63]. Water interaction with the donor and acceptor atoms of each ligand were calculated at HF/6 31G(d) level of theory to determine charges. The target data for angles and dihedral angles involved computing Hessians and dihedral angle potential energy scans at MP2/6–31G(d) level of theory, as per standard ffTK parameterization procedure. The Gaussian 16 software [64] was used to obtain the necessary QM data. The time step for MD simulation was set to 2 fs and the trajectory was saved every 1 ps. Long-range electrostatic interactions were calculated using the Particle Mesh Ewald (PME) method and a cut-off distance of 10 Å. Temperature control was achieved using Langevin dynamics and a Langevin piston was used for the pressure control.

As a starting point for MD simulations, we have chosen binding modes with the highest score that were involved in the interactions with critical residue *Glu-*30. The enzyme in the unliganded state (DHFR) was also taken into consideration, to compare to the protein–ligand complexes and to measure the impact of five selected inhibitors. To reflect the human cell environment, each system was prepared for a simulation by emerging in the explicit water solvent box (TIP3P) and applying periodic boundary conditions. The dimensions of the box were chosen, so that there is a layer of water 10 Å wide in each direction. All residues were set to their proper protonation state in physiological pH = 7.4 and 0.15 M of NaCl was added using Autoionize plugin in VMD software [65]. Then, each system was minimized for 10,000 steps using conjugate gradient algorithm and gradually heated from 0 to 310 K with 2k increments every 0.5 ps. Before the final simulation, all systems were equilibrated for 1 ns under isothermal-isobaric (NVT) ensemble. The production run used for analysis was carried out under isothermal–isochoric (NPT) ensemble for 20 ns. VMD was used to visually inspect the trajectory and to extract the necessary data using Tcl scripts for analysis, i.e., Root Mean Square Deviation (RMSD), Root Mean Square Fluctuation (RMSF), Solvent Accessible Surface Area (SASA), and Radius of Gyration (Rg).

Root Mean Square Deviation (RMSD) is the most commonly used quantitative measure of the similarity between two superimposed atomic coordinates. RMSD values are given in Å and calculated using Equation (1):(1)$$\mathrm{RMSD}=\sqrt{\frac{1}{n}\overset{n}{\underset{i\;=\;1}{\sum }}d_{i}^{2}}$$
where the averaging is performed over n pairs of equivalent atoms and d_i_ is the distance between the two atoms in the i-th pair. In this study, RMSD was calculated in relation to the NVT equilibrated structure for Cα atoms of all residues over 20 ns MD simulation.

The root-mean-square fluctuation (RMSF) measures the average deviation of a residue over simulation time from a reference position. Thus, RMSF shows the portions of a protein structure that are flexible or rigid during the simulation. In this study, we took into consideration all atoms within the residue except hydrogens for RMSF analysis
(2)$$R_{g}=\sqrt{\overset{n}{\underset{i=1}{\sum }}\frac{m_{i}s_{i}^{2}}{\sum_{i=1}^{n}m_{i}}}$$

For a macromolecule composed of *n* atoms, of masses *m_i_*, *i* = 1, 2,..., *n* located at fixed distances *s_i_* from the centre of mass, the radius of gyration is the square-root of the mass average of *s_i_*^2^ over all atoms Equation (2) [66]. It is an indicator of protein structure compactness [67] and serves as an estimation of how secondary structures are compactly packed in the protein.

Solvent accessible surface area (SASA) is defined as the surface characterized around a protein by a hypothetical center of a solvent sphere with the van der Waals contact surface of the molecule [68]. It reflects the expansion of the protein and may indicate protein folding. A typical value for a water solvent of 1.4 Å was set for probe radius.

## 4. Conclusions

A series of novel TMP analogs **1**–**18** containing an amide bond was synthesized and investigated. Compounds **13**–**14** and **17**–**18** were characterized by a higher binding strength to *p*BR322 plasmid. The determination of values of association constants of drug–DNA complexes assay revealed that all compounds can bind to the studied DNAs. These data indicated that compounds **1**–**18** interacted with AT as well as GC-base pairs and we can observe the greatest preference for AT-base pairs of compound **14** and for GC-pairs of **3**. Compound **18** showed high-value binding constants for T4 coliphage DNA and confirmed their minor-groove selectivity. The in vitro experimental findings revealed that all the newly designed and synthesized compounds, especially **2**, **6**, **13**–**14**, and **16**–**18**, exhibited higher activity against the DHFR enzyme and higher binding affinity than standard TMP.

The results obtained from theoretical calculations show that there is a considerable attraction between our inhibitors and the catalytically vital Glu-30. Among them, five were determined to be particularly effective, namely **2**, **6**, **13**, **14,** and **16**. Detailed analysis of their impact on the enzyme was carried out using data from MD simulation: RMSD, RMSF, SASA, and Rg (Figure 7). Each of the investigated molecules were found to lower RMSD as compared to the apo-protein. The most substantial stabilization was observed for DHFR and **13** complexes, which remained low values of RMSD and small fluctuations for the entire time. On the other hand, RMSF examination showed that derivative **2** caused the least fluctuations, lowering this value for almost the entire sequence. That is contrary to the effect of molecules **6** and **14**, which increased flexibility significantly for certain regions. SASA and Rg results indicated that protein was the most compact in an unliganded state, although deviations from the values of DHFR were marginal. Compound **2** formed the most stable connection with Glu-30, though in general, compound **6** formed the most H-bonds (Figure 9).

The introduction of an amide bond into the newly synthesized TMP analogs increased their affinity to human DHFR compared to unmodified TMP (−7.5 kcal/mol) (Table 1). This was also validated by our MD study, where we found that Ala-9, Val-115, and Tyr-121 residues were responsible for the stabilization of our ligands by interacting with the amide group. Interaction with Phe-34 residue was also deemed important, as it was interacting via t-shaped π–π stacking with aromatic moiety that binds to the Glu-30 catalytic residue.

In summary, these results confirmed our assumption about synthesizing multi-target compounds: the DNA binding effect and DHFR inhibitory activity, which are proved by molecular docking studies. These structures may have an interesting future as a template for developing new analogs with potential anticancer properties. We plan to do further in vitro investigations of the activity on cancer cell lines to confirm their effectiveness and potential use in therapeutic applications.

## Data Availability

Not applicable

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
