# Peer review of "Synthesis, Biological Activity, and Molecular Dynamics Study of Novel Series of a Trimethoprim Analogs as Multi-Targeted Compounds: Dihydrofolate Reductase (DHFR) Inhibitors and DNA-Binding Agents"

_ijms, 2021, doi:10.3390/ijms22073685_

Round 1
Reviewer 1 Report
Interesting paper by Wrobel and coworkers
The manuscript is well written, clearly described, the research well planned and executed. The results are sounding and the experimental section well specified.
I see, however, some points which could be modified to improve the quality:
1- Discuss (may be briefly, a paragraph) in the text , after the introduction, the synthetic access to the new compounds. As only details can be found in the experimental section. In this regard, why authors use TBTU as coupling agent for the interaction of an acyl chloride with an amine? It is normally required for the direct amidation of an acid. The acid chloride is already activated! Please explain as it is not intuitive, and furthermore, at first sight does not make any mechanistic sense. Could it be a mistake (the coupling being done with the acid itself?)
2- Very important. Add in the supporting information section, the copies of the 1H and 13 C NMR spectra of the new compounds. This would be the guarantee of the purity of the synthesized products.
3- There is one sentence that I can not understand. Please, rewrite or clarify.
Page 3 " In the work of Wan et al. [37] the authors used neutron and ultrahigh-resolution X-ray crystallography to establish the catalytic activity of E. coli." Does it refer to the E coli enzyme?
Author Response
Thank you for your remarks.
The revised manuscript contains answers to your comments. We hope that these minor revisions were made to the text will allow the work to be published quickly in International Journal of Molecular Sciences
We are (main author and co-author) a very grateful for the effort of reviewing the manuscript. The manuscript was carefully polish to make the results of this study clear to the readers. We hope that the changes introduced will be recognized as modifications that significantly improve the quality and understanding of the review.
- English language and style are fine/minor spell check required
Response: English language and style were improved. Green indicates corrections.
- Discuss (may be briefly, a paragraph) in the text , after the introduction, the synthetic access to the new compounds. As only details can be found in the experimental section.
Response: We add the synthetic access to the new compounds in paragraph:
,,For the TMP derivatives preparation procedure aromatic amino-nitro compounds A1-9 and selected acid chlorides E1-5 were used as substrates to obtain 18 novel TMP analogs containing an amide bond, as shown in Figure 4. Compounds with the structure II were obtained according to the reported procedure [46] from p-nitrophenyl carbonate Wang resin I, as shown in Scheme 1. After grafting the nitroamines to the resin, reduction of the nitro group of structure II was carried out using tin (II) chloride dihydrate in DMF. Acylation of 1-9 resin-bound amines III, using E1-5 chlorides in the presence of DMAP in methylene chloride at room temperature produced the resin-bound nitro compounds with structure IV. Cleavage by 95% trifluoroacetic acid in dichloromethane gave a satisfactory yeld of the desired compounds with structure V. The structures of compounds A1-9, acid chlorides E1-5, the analytical and spectrometric data are presented in Table 3(Supplementary Materials).,,
- In this regard, why authors use TBTU as coupling agent for the interaction of an acyl chloride with an amine? It is normally required for the direct amidation of an acid. The acid chloride is already activated! Please explain as it is not intuitive, and furthermore, at first sight does not make any mechanistic sense. Could it be a mistake (the coupling being done with the acid itself?)
Response: It was a incorrectly written reaction step. We used DMAP in acylation step, not TBTU. The acid chloride was already activated.
- Very important. Add in the supporting information section, the copies of the 1H and 13 C NMR spectra of the new compounds. This would be the guarantee of the purity of the synthesized products.
Response: We add in the supporting information section, the copies of the 1H and 13 C NMR spectra of the new compounds.- Supplementary materials.
PoczÄ…tek formularza
- There is one sentence that I can not understand. Please, rewrite or clarify.
Page 3 " In the work of Wan et al. [37] the authors used neutron and ultrahigh-resolution X-ray crystallography to establish the catalytic activity of E. coli." Does it refer to the E coli enzyme?
,,In the work of Wan et al. [37] the authors used neutron and ultrahigh-resolution X-ray crystallography to establish the catalytic activity of the E. coli DHFR enzyme (ecDHFR). ,,

Reviewer 2 Report
This is a very interesting paper providing significant results on the novel series of Trimethoprim analogues as multi-targeted compounds. I read this paper with great interested and authors have discussed their computational modeling results with their experimental work in details.
However, I would strongly suggest authors to add previous molecular modeling work on this subject done by them or other researchers in the introduction of this paper.
It is of fundamental interest, and should hence be published after addressing above comments.
Author Response
Thank you for your remarks.
The revised manuscript contains answers to your comments. We hope that these minor revisions were made to the text will allow the work to be published quickly in International Journal of Molecular Sciences
We are (main author and co-author) a very grateful for the effort of reviewing the manuscript. The manuscript was carefully polish to make the results of this study clear to the readers. We hope that the changes introduced will be recognized as modifications that significantly improve the quality and understanding of the review.
- English language and style are fine/minor spell check required
Response: English language and style were improved. Green indicates corrections.
- However, I would strongly suggest authors to add previous molecular modeling work on this subject done by them or other researchers in the introduction of this paper.
Response:
The highlighted fragment shows previous molecular modeling work on this subject done by other researchers:
,,Algul et al.[28] developed a new nonclassical series of propargyl-linked DHFR inhibitors. It was observed that interactions of propargyl-linked inhibitor (compound c) (Figure 2) with Leu22, Thr56, Ser59, Ile60 could potently inhibit human DHFR (hDHFR) in contrast to the weak inhibition of hDHFR by TMP.,,
